# Prognostic Impact of CD36 Immunohistochemical Expression in Patients with Muscle-Invasive Bladder Cancer Treated with Cystectomy and Adjuvant Chemotherapy

**DOI:** 10.3390/jcm11030497

**Published:** 2022-01-19

**Authors:** Juan Carlos Pardo, Tamara Sanhueza, Vicenç Ruiz de Porras, Olatz Etxaniz, Helena Rodriguez, Anna Martinez-Cardús, Enrique Grande, Daniel Castellano, Miquel A. Climent, Tania Lobato, Lidia Estudillo, Mireia Jordà, Cristina Carrato, Albert Font

**Affiliations:** 1Medical Oncology Department, Catalan Institute of Oncology, Ctra. Can Ruti- Camí de les Escoles s/n, 08916 Badalona, Spain; jcpardor@iconcologia.net (J.C.P.); oetxaniz@iconcologia.net (O.E.); 2Catalan Institute of Oncology, Badalona Applied Research Group in Oncology (B·ARGO), 08916 Badalona, Spain; vruiz@igtp.cat (V.R.d.P.); amartinezc@igtp.cat (A.M.-C.); tlobato@igtp.cat (T.L.); 3Germans Trias i Pujol Research Institute (IGTP), 08916 Badalona, Spain; hrodriguez@igtp.cat (H.R.); mjorda@igtp.cat (M.J.); 4Pathology Department, Hospital Universitari Germans Trias i Pujol, 08916 Badalona, Spain; tsanhueza.germanstrias@gencat.cat (T.S.); ccarrato.germanstrias@gencat.cat (C.C.); 5Medical Oncology Department, Hospital Universitario Ramon y Cajal, 28034 Madrid, Spain; egrande@oncomadrid.com; 6Medical Oncology Department, MD Anderson Cancer Center, 28033 Madrid, Spain; 7Medical Oncology Department, Hospital Universitario 12 de Octubre, 28041 Madrid, Spain; cdanicas@hotmail.com; 8Medical Oncology Department, Instituto Valenciano de Oncologia, 46009 Valencia, Spain; macliment@fivo.org; 9Genetic and Molecular Epidemiology Group, Spanish National Cancer Research Centre (CNIO), CIBERONC, 28029 Madrid, Spain; lestudillo@cnio.es

**Keywords:** CD36, MIBC, bladder cancer, adjuvant chemotherapy, prognostic biomarker, lipid metabolism, fatty acid

## Abstract

Neoadjuvant chemotherapy followed by a cystectomy is the standard treatment in muscle-invasive bladder cancer (MIBC). However, the role of chemotherapy in the adjuvant setting remains controversial, and therefore new prognostic and predictive biomarkers are needed to improve the selection of MIBC patients. While lipid metabolism has been related to several biological processes in many tumours, including bladder cancer, no metabolic biomarkers have been identified as prognostic in routine clinical practice. In this multicentre, retrospective study of 198 patients treated with cystectomy followed by platinum-based adjuvant chemotherapy, we analysed the immunohistochemical expression of CD36 and correlated our findings with clinicopathological characteristics and survival. CD36 immunostaining was positive in 30 patients (15%) and associated with more advanced pathologic stages (pT3b-T4; *p* = 0.015)**.** Moreover, a trend toward lymph node involvement in CD36-positive tumours, especially in earlier disease stages (pT1-T3; *p* = 0.101), was also observed. Among patients with tumour progression during the first 12 months after cystectomy, disease-free survival was shorter in CD36-positive tumours than in those CD36-negative (6.51 months (95% CI 5.05–7.96) vs. 8.74 months (95% CI 8.16–9.32); *p* = 0.049). Our results suggest an association between CD36 immunopositivity and more aggressive features of MIBC and lead us to suggest that CD36 could well be a useful prognostic marker in MIBC.

## 1. Introduction

Bladder cancer (BC) is the tenth most common cancer, and its incidence is steadily rising worldwide, with the highest rates in developed countries [1,2]. Most BCs are diagnosed in early stages of the disease and present only superficial growth and scarce infiltration to the subepithelial connective tissue. However, roughly 25% of BCs invade beyond the muscularis propria and are referred to as muscle-invasive bladder cancer (MIBC), with the potential to spread directly to adjacent pelvic structures or develop lymphatic and hematogenous metastases, which are associated with poor prognosis [3]. In addition to other well-known variables, obesity has recently come to light as a risk factor for bladder cancer [4]. Although the underlying reasons for the association of obesity with bladder cancer risk are not well understood, several explanations have been proposed, including insulin resistance with increased production of insulin-like growth factor 1 and elevated nutrient availability [5], reprogramming of cellular lipid metabolism [6], chronic inflammation associated with obesity, and the role of adipocytes in the tumour microenvironment [7].

The cornerstone of treatment for non-MIBC (NMIBC) is transurethral resection of the bladder (TURB) with or without Bacillus Calmette-Guérin (BCG) instillation or chemotherapy [8]. On the other hand, MIBC patients benefit more from radical cystectomy (RC) with extended lymphadenectomy [9]. Nevertheless, between 10 and 20% of NIMBC patients with high-risk features will relapse or progress to MIBC, and nearly 50% of patients with MIBC will develop metastatic disease after undergoing RC. The goal of perioperative chemotherapy is to treat micrometastatic disease and thus avoid patient relapse. However, the role of adjuvant chemotherapy remains controversial [10,11,12,13,14,15]. Clinical guidelines recommend a cisplatin-based regimen for patients with high-risk MIBC (pT3–pT4 and/or lymph node involvement) who are fit for this treatment and who have not received neoadjuvant chemotherapy [12,16].

Prognosis in MIBC is generally based on pathological stage after RC, and patients with greater depth of tumour invasion and positive lymph nodes have a shorter five-year overall survival (OS) [17]. Nevertheless, there is a clear need for novel biomarkers to identify MIBC patients who are less likely to benefit from the standard treatment and who can be offered alternative and potentially more effective therapies. In addition, in NMIBC, it could also be relevant to determine which patients may benefit from standard treatment as well as new therapeutic strategies such as immunotherapy [18]. This situation is now changing thanks to our better understanding of the molecular biology of BC, which has allowed us to identify new prognostic and predictive biomarkers that could be used in the future. In this context, several classifications of MIBC based on mRNA expression have defined intrinsic subtypes that are associated with OS and with specific clinicopathological characteristics [19,20,21,22,23,24]. Alterations in lipid metabolism are emerging as an important prognostic factor in several tumours, and potential new therapies against these dysregulated anabolic and catabolic pathways have been suggested [25]. These translational breakthroughs need to be validated and introduced into routine clinical practice if we are to move towards personalized medicine in MIBC [26].

Several studies have proposed CD36 expression as an unfavourable prognostic factor for different tumour types [27,28,29]. CD36 is a multifunction transmembrane glycoprotein in the class B scavenger receptor family (also known as a fatty acid (FA) translocase). CD36 is involved in long-chain FA uptake, as well as in tumour immuno-editing, metastasis, treatment response, and angiogenesis through its association with different ligands [30]. In the present study, we have analysed the immunohistochemical expression of CD36 in a retrospective cohort of MIBC patients treated with RC followed by adjuvant chemotherapy and correlated our findings with patient clinicopathological characteristics and outcomes.

## 2. Materials and Methods

### 2.1. Study Population

We retrospectively included 198 patients with MIBC treated between 2000 and 2014 at four Spanish centres. Inclusion criteria were histologically confirmed muscle-invasive urothelial bladder carcinoma treated by RC with negative surgical margins. In addition, patients had to have received at least one cycle of adjuvant chemotherapy and have sufficient available viable tumour tissue to be included in a tissue microarray (TMA).

Data were collected from medical records by physicians using the REDCap system (a web-based EDC system that stores and audits information centrally and securely in accordance with the legal requirements regarding data protection). Information was collected on diagnostic procedures, disease stage, tumour characteristics, initial treatment, date of last control, date and characteristics of progression, and date and cause of death.

### 2.2. Tissue Specimens

Two diagnostic blocks of formalin-fixed paraffin-embedded (FFPE) tissue were retrieved for each patient. Haematoxylin-eosin-stained sections were assessed by two expert pathologists (TS and CC) who corroborated the diagnosis and selected representative tumour areas. Three 0.6-mm cylindrical tissue cores were extracted from each sample and inserted into eight TMA recipient blocks using an MTA-1 TMA workstation (Beecher Instruments, Silver Spring, MD, USA). Cores obtained from each patient were inserted in distant, randomly assigned positions of the recipient blocks to rule out bias in the evaluation. TMA sections were used to determine expression of CD36 by immunohistochemistry (IHC).

### 2.3. CD36 Immunostaining

CD36 immunostaining (clone HPA2018, 1:100 dilution, Sigma-Aldrich, St. Louis, MO, USA) was performed on 4-μm-thick sections, using hepatic tissue as a positive control. Results were evaluated by two pathologists (TS and CC). CD36 immunoreactivity was evaluated semi-quantitatively based on the percentage of neoplastic cells with positive membrane with or without added cytoplasmatic expression as follows: 0 (<1% of positive tumor cells), 1 (1–5%), 2 (5–25%), 3 (25–50%), 4 (50–75%), and 5 > 75%). Samples that showed CD36 immunoreactivity in at least 1% of neoplastic cells (scores 1 to 5) were considered CD36-positive. For further analysis, in the subgroup of patients with lymph node metastasis from Hospital Universitari Germans Trias i Pujol, whole sections (4-μm-thick) from cystectomy and lymph node specimens were stained with CD36 antibody and analysed.

### 2.4. Ethical Issues

Ethical approval for retrospective use of available samples and information in this study was obtained from the Institutional Review Board of each participating centre prior to study onset. We applied the Spanish data protection law 15/13 December 1999 (Real Decreto 1720/21 December 2007) as well as the Spanish Biomedical Research law 14/3 July 2007 (Real Decreto 1716/18 November 2011) on the protection of individuals with regard to the processing of personal data and biosamples and on the free movement of such data and material.

### 2.5. Statistical Analyses

Continuous variables were summarized as means, medians, and standard deviations, and categorical variables as percentages. Bivariable associations for categorical variables were evaluated by Chi-square or Fisher’s exact test. OS was defined as the time from cystectomy to death from any cause and disease-free survival (DFS) as the time from cystectomy to disease relapse. OS and DFS were analysed both in the entire patient cohort and in the sub-group of patients with disease progression during the first 12 months after RC. Survival curves were derived using the Kaplan–Meier method and compared with log-rank tests between patients with CD36-positive and CD36-negative tumours. To study the role of each covariable as prognostic factors in survival, hazard ratios (HR) and their 95% confidence intervals (95% CI) were estimated using univariate Cox proportional hazards regression models, considering as statistical significant *p*-values under 0.05. All statistical analyses were performed using SPSS software version 16.0 (SPSS Inc., Chicago, IL, USA).

## 3. Results

### 3.1. Patients

We originally selected a total of 257 patients from the four hospitals, 50 of whom were excluded from the study since they did not meet the inclusion criteria. An additional nine patients were not included in the final analysis since the IHC results were not informative due to technical issues with immunostaining of the samples. The remaining 198 patients were included in further analyses (Figure 1).

Patient clinicopathological characteristics are shown in Table 1. Median age at diagnosis was 65 years (range, 41.3–92.6), and 90% of patients were male. Histology was pure urothelial carcinoma in 167 cases (85%), while in 31 cases (15%), divergent differentiation (squamous or glandular) or special variant pattern (micropapillary or lymphoepithelioma-like) was seen. According to AJCC 8th edition pathologic stage classification: 6 cases (3%) were diagnosed as stage II, 163 cases (82%) were stage III, and 14 cases (8%) stage IV; in 15 cases (7%) pathologic stage could not be correctly determined. Adjuvant chemotherapy was cisplatin-based in 67% of patients and carboplatin-based in 32%. The median number of cycles was four and 13% of patients completed six or more cycles. There were no significant differences in clinicopathological characteristics between groups according to CD36 expression (Table 1).

### 3.2. CD36 Immunostaining

Tumour samples from 30 patients were CD36-positive, with either a membrane or a membrane and cytoplasm pattern (Figure 2). Since heterogeneous immunoreactivity was seen in the TMA within the same spot or between different spots of the same sample, a three-spot average was given for every CD36-positive case and whole sections from five cystectomy specimens were immunostained for further evaluation. In the whole sections, CD36 staining again showed intratumoral heterogeneity, with positive and negative areas, but the percentage of the total positive tumoral cells was equivalent to that obtained in the TMA (same score). There were no differences in the percentage or intensity of CD36 expression between superficial and deeper tumour portions. As expected, adipocytes in the perivesical fat tissue were immunopositive for CD36 (Figure 3A–C).

We also analysed whole sections of metastatic lymph nodes from six CD36-positive and seventeen CD36-negative cases. When cystectomy specimens were CD36-negative, the paired lymph node specimens were also CD36-negative, and when cystectomy specimens were CD36-positive, the paired lymph node specimens were also CD36-positive, with similar immunostaining characteristics (heterogeneous, with a similar percentage of positive cells and the same score) (Figure 3D–F).

One of the TMA cases showed an unusual dot-like additional pattern that was also demonstrated in metastatic lymph node deposits (Figure 2D).

### 3.3. Correlation of CD36 Expression and Pathological Staging

The distribution of patient characteristics was well-balanced between patients with CD36-positive and CD36-negative tumours (Table 1), including pathological stage, histology, and type of chemotherapy. There was a difference between patients with CD36-positive and CD36-negative tumours in the depth of tumour invasion (pT stage), with more frequent CD36-positivity in more advanced tumours (pT3b-T4) (*p* = 0.015). In addition, there was a trend toward a greater involvement of pathological lymph nodes in CD36-positive cases, especially among those with earlier stage disease (pT1-T3) (*p* = 0.101) (Figure 4). The Cancer of the Bladder Risk Assessment (COBRA) score [31] is a simplified risk stratification tool for estimating cancer-specific survival after RC that accurately incorporates the relative contribution of clinical characteristics, such as age, primary T stage and pathologic lymph node density (number of affected nodes/number of nodes removed). When we examined the potential relationship between CD36 immunopositivity and the COBRA score, we found no differences between CD36-positive and CD36-negative cases.

### 3.4. CD36 and Patient Outcomes

There were no differences in type of adjuvant therapy administered (cisplatin-based vs carboplatin-based) between CD36-positive and CD36-negative cases, and the rate of disease progression after RC was also similar, both overall and when adjusted by chemotherapy regimen. There were no differences between the two groups in radiographic treatment response to first-line adjuvant chemotherapy. Of the 121 patients who progressed, 81 (67%) received second-line treatment: 57% platinum-based chemotherapy, 37% taxanes, and 6% vinflunine or other agents.

With a median follow-up of 43.76 months (range, 2.76–182.34), for all 198 patients, patient clinicopathological characteristics were reviewed to identify significant prognostic determinants. By univariate cox regression analysis performed on survival, advanced pathologic stage (pT3b-T4), lymph node involvement, high COBRA score (4–7), and lymph vessel invasion emerged as significant worse prognostic determinants, while the use of cisplatin chemotherapy was found to be a protective factor against the use of carboplatin, consistent with published literature [32,33]. However, there were no differences in OS or DFS between CD36-positive and CD36-negative patients, suggesting that CD36 could not be considered a prognostic factor in this population (Table 2). Focussing on the subgroup of patients who presented worse prognosis in our cohort, identified as those who progressed during the first 12 months after RC, CD36-positive patients had shorter DFS than CD36-negative patients (6.51 months [95% CI 5.05–7.96] vs. 8.74 months [95% CI 8.16–9.32], respectively; *p* = 0.049) (Figure 5).

## 4. Discussion

Altered energy metabolism is one of the emerging hallmarks of cancer described by Hanahan and Weinberg [34]. However, inhibitors of this process have not been explored in depth over the last 50 years. Fatty acid (FA) metabolism is an important part of lipid metabolism and participates in the homeostasis of tumour cells, enabling different processes for tumour survival, maintaining tumour growth and proliferation, fulfilling energy requirements, and providing metabolites for anabolic processes [35]. FA metabolism involves the uptake of exogenous FAs, lipid transport and storage and *de novo* synthesis of FAs. CD36 is a multifunctional transmembrane receptor that is involved in FA uptake, innate immunity, and angiogenesis [30,36]. CD36-positive metastasis-initiating cells were associated with poor prognosis, suggesting that it may be a potential prognostic biomarker [27]. Recently, the prognostic value and immunological role of CD36 has been evaluated in multiple tumors and paratumoral tissue. The results indicate that, in fact, the influence of CD36 on patient prognosis depends on the type of cancer [37].

In the present study, we analysed CD36 by IHC in TMAs from tumour specimens obtained during cystectomy and found that CD36 immunopositivity was significantly associated with greater depth of tumour invasion (pT3b-pT4 stage) and showed a trend toward association with greater lymph node involvement (pN stage) (*p* = 0.391), both of which are well-known markers of poor prognosis in MIBC [38]. Several studies have reported a differential effect on survival between microscopic invasion of perivesical tissue (pT3a) and macroscopic/extravesical mass (pT3b) [39,40], leading us to speculate that the CD36 positivity that we have observed in tumours invading the perivesical fat or peritoneum may reflect a metabolic change that enables cancer cells to obtain the energy and nutrients needed to confront metabolic stress and maintain tumour progression [41,42].

CD36 has been related to metastasis development and lymph node invasion [28], and CD36 inhibition led to a significant reduction in the number and size of pathological lymph nodes in murine models [27]. Our finding of a trend toward association of CD36 immunopositivity and lymph node involvement, which was strongest in earlier stage disease (pT1-T3), suggests that CD36 could be involved in an initial cellular change to a migratory phenotype [43].

These findings suggest that upregulation of CD36 might be an important contributory factor to MIBC progression and could be used to identify patients with poor prognosis who may also benefit from a more extensive lymphadenectomy or need a different treatment approach instead of cystectomy followed by adjuvant chemotherapy [29]. These patients could be offered neoadjuvant chemotherapy [44,45] or an alternative adjuvant regimen, such as immunotherapy [46].

Our study failed to demonstrate an association between CD36 immunopositivity and shorter DFS or OS in the entire cohort. However, when we focused on the subgroup of patients with a DFS of less than 12 months, we found that DFS for patients with CD36-positive tumours was two months shorter than for those with CD36-negative tumours (6.51 vs. 8.74 months; *p* = 0.049), indicating that CD36 positivity was linked to more aggressive tumours. We can suggest that in clinical practice, patients with CD36-positive tumours should be closely followed to detect early progression and offer effective alternative treatments as soon as possible [47,48,49].

To date, few studies have explored the impact of CD36 in urothelial carcinoma. Bowden and colleagues performed a transcriptome study in high-risk non-muscle- invasive micropapillary BC and found that high CD36 expression correlated with a trend to shorter time to progression. They generated a prognostic risk index of three genes (CD36, FABP3 and RAETE1) that was significantly associated with shorter time to progression [50]. In addition, Jeong and colleagues recently carried out a study in non-MIBC examining by IHC the protein expression of FATP4, CD36 and ACSL1, all of which are involved in lipid metabolism. They found that CD36 was linked to higher-grade tumours and more advanced disease stage, but not to survival. However, they conclude that the three genes might promote progression and could be used for risk stratification [51].

To the best of our knowledge, this is the first study to analyse the role of CD36 in MIBC by IHC, a rapid, simple, and cost-effective procedure that can be easily implemented in routine clinical practice. Our findings suggest that CD36 immunopositivity may be a marker of poor prognosis since it was linked to greater pT and pN stages as well as to extremely short DFS in the subgroup of patients with more aggressive disease. However, we are aware of the limitations of our study, including its retrospective nature, the limited number of patients included, and the heterogeneity of expression observed in the TMAs, and we suggest that the impact of CD36 warrants further investigation. If our results are validated prospectively, CD36 analysis by IHC could be a useful clinical tool for determining the prognosis of MIBC patients.

## Figures and Tables

**Figure 1 jcm-11-00497-f001:**
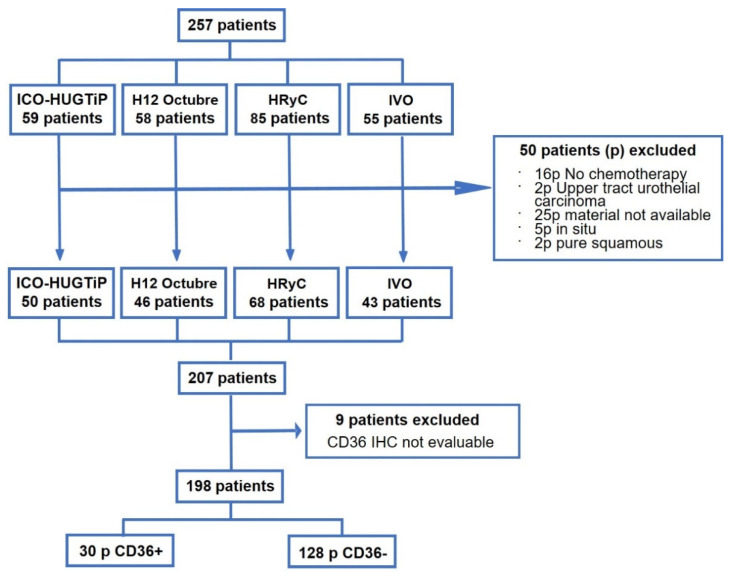
Flow chart showing patient inclusion in the study. ICO-HUGTiP, Institut Català d’Oncologia-Hospital Universitari Germans Trias i Pujol (Badalona, Barcelona); H12 Octubre, Hospital 12 de Octubre (Madrid); HRyC, Hospital Ramon y Cajal (Madrid); IVO, Instituto Valenciano de Oncología (Valencia).

**Figure 2 jcm-11-00497-f002:**
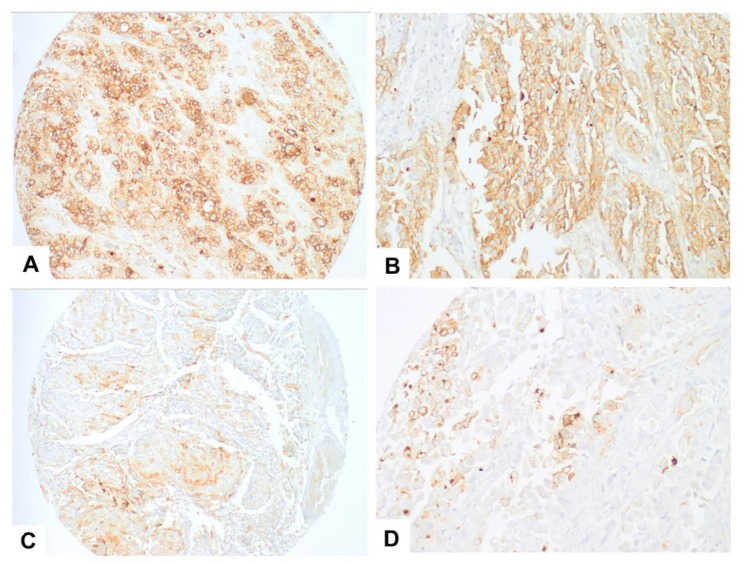
CD36 immunostaining in TMA samples. Most cases showed a diffuse linear membranous positivity with some cytoplasmic positivity as shown in these images from two different cases ((**A**): 10×; (**B**): 20×). In some tumours, immunoreactivity was heterogeneous (with positive and negative tumoral cells) ((**C**): 10×), and in a single case, we saw an additional dot-like pattern ((**D**): 20×).

**Figure 3 jcm-11-00497-f003:**
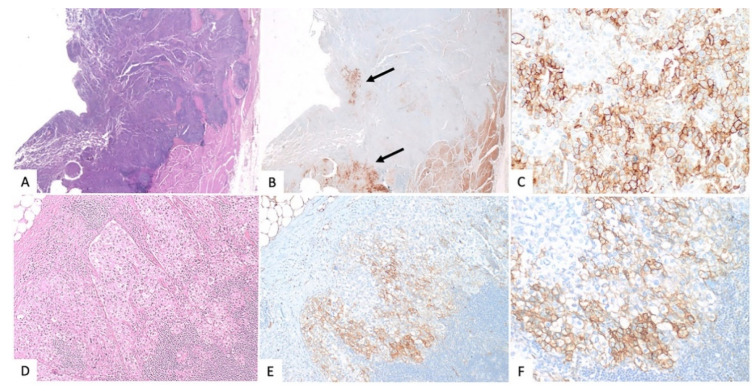
CD36 immunoreactivity in whole sections. Heterogenous positivity was evident in both cystectomy ((**A**): H-E 4×; (**B**): CD36 4×) and lymph node whole sections ((**D**): H-E 20×; (**E**): CD36 20×) with membranous positivity with/without additional cytoplasmic positivity ((**C**,**F**), 40×). There were positive and negative tumoral areas in both superficial and deep parts of the tumour in the bladder wall (arrows in (**B**)) (All of these images are from the same case).

**Figure 4 jcm-11-00497-f004:**
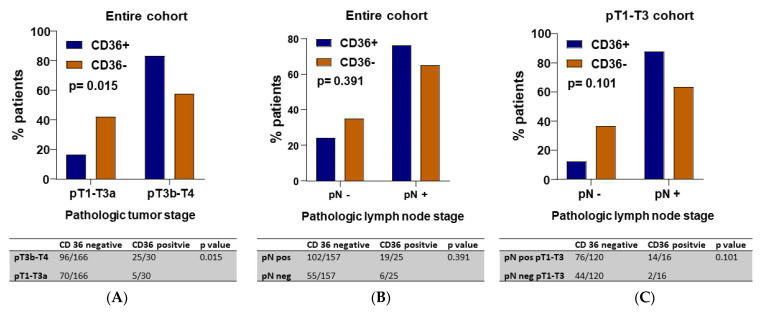
CD36 immunopositivity was significantly associated with greater depth of tumour invasion (pT3b-pT4 stage) (*p* = 0.015) (**A**) and showed a trend toward association with greater lymph node involvement (pN stage) (*p* = 0.391) (**B**), which was strongest in patients with earlier stage disease (pT1-T3) (*p* = 0.101) (**C**).

**Figure 5 jcm-11-00497-f005:**
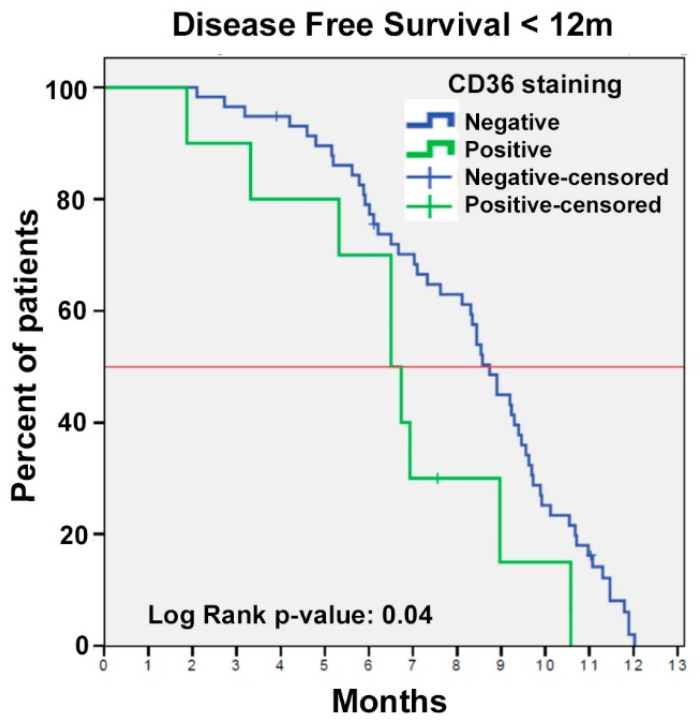
Kaplan Meier survival curve in the subgroup of patients with worse prognosis, identified as those who progressed during the first 12 months after RC, CD36-positive patients had shorter median PFS than CD36-negative patients (6.51 months (95% CI 5.05–7.96) vs 8.74 months (95% CI 8.16–9.32), respectively; (*p* = 0.049).

**Table 1 jcm-11-00497-t001:** Patient clinicopathological characteristics.

Characteristic	All Patients N = 198 N (%)	CD36 Immunostaining		
		CD36-Negative N = 168 N (%)	CD36-Positive N = 30 N (%)	*p* *
Sex				0.1
Male	177 (90)	153 (91)	24 (80)	
Female	21 (10)	15 (9)	6 (20)	
Age				0.6
<65 years	92 (46)	79 (47)	13 (43)	
>65 years	105 (53)	88 (52)	17 (57)	
NA	1 (1)	1 (1)	0 (0)	
Histology				0.4
Pure urothelial	167 (85)	140 (83)	27 (90)	
Others	31 (15)	28 (17)	3 (10)	
Tumour invasion				0.3
pT1	1 (1)	1 (1)	0 (0)	
pT2	22 (11)	19 (11)	3 (10)	
pT3	122 (62)	106 (63)	16 (53)	
pT4	53 (27)	42 (25)	11(37)	
Nodal status (pN)				0.7
pN0	61 (31)	55 (33)	6 (20)	
pN1	47 (25)	39 (23)	8 (27)	
pN2	60 (31)	51 (30)	9 (30)	
pN3	14 (7)	12 (7)	2 (7)	
NX	16 (8)	11 (7)	5 (16)	
AJCC Stage (8th edition)				0.5
II	6 (3)	6 (4)	0 (0)	
IIIA	99 (50)	85 (51)	14 (47)	
IIIB	64 (32)	55 (33)	9 (30)	
IVA	9 (5)	7 (4)	2 (7)	
IVB	5 (3)	4 (2)	1 (3)	
NA	15 (7)	11 (6)	4 (13)	
Adjuvant chemotherapy				0.6
Carboplatin-based	64 (32)	56 (33)	8 (27)	
Cisplatin-based	133 (67)	111 (66)	22 (73)	
Other	1 (1)	1 (1)	0 (0)	
Progressive disease				0.2
Yes	121 (61)	106 (63)	15 (50)	
No	69 (35)	56 (33)	13 (43)	
NA	8 (4)	6 (4)	2 (7)	

* *p*-value for comparison between CD36-positive and CD36-negative patients, calculated with Chi-square or Fisher’s exact test as appropriate. AJCC, American Joint Committee on Cancer; NA, not available.

**Table 2 jcm-11-00497-t002:** Prognostic factors in the entire cohort.

	Disease Free Survival (DFS)		Overall Survival (OS)	
Prognostic Factor	HR (CI 95%)	*p* Value	HR (CI 95%)	*p* Value
Sex				
Male vs. female	0.77	0.355	0.88	0.647
	(0.44–1.34)		(0.50–1.53)	
Age				
≥65 vs. <65 years	1.2	0.321	1.22	0.264
	(0.84–1.73)		(0.86–1.74)	
Pathologic Stage:				
pT3b-T4 vs. pT1-T3a	1.73	0.006 **	1.58	0.016 *
	(1.17–2.56)		(1.09–2.30)	
pN+ vs. pN−	1.78	0.010 *	1.74	0.009 **
	(1.15–2.76)		(1.14–2.63)	
Lymph vessel invasion				
Present vs. absent	2.58	0.001 **	2.33	0.001 **
	(1.50–4.45)		(1.40–3.88)	
Histology				
Pure urothelial vs. others	1.29	0.278	1.49	0.083
	(0.81–2.06)		(0.95–2.35)	
COBRA score				
4–7 vs. 0–3	1.43	0.061	1.43	0.048 *
	(0.98–2.08)		(1.00–2.05)	
Adjuvant chemotherapy:				
Cisplatin vs. carboplatin	0.61	0.009 **	0.68	0.036 *
	(0.42–0.88)		(0.48–0.98)	
CD36 immunostaining				
CD36+ vs. CD36−	0.78	0.357	0.88	0.62
	(0.45–1.33)		(0.54–1.44)	
Progression ≤ 12 months				
Yes vs. No	NA	NA	11.36	<0.001 **
			(7.38–17.49)	

Hazard ratios for disease-free survival and overall survival calculated by Cox regression univariate analysis. * Statistically significant *p* ≤ 0.05. ** Statistically significant *p* ≤ 0.01. NA, not available.

## Data Availability

Data from this study are available upon reasonable request to the corresponding author.

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
