# Peer review of "Prognostic Impact of CD36 Immunohistochemical Expression in Patients with Muscle-Invasive Bladder Cancer Treated with Cystectomy and Adjuvant Chemotherapy"

_jcm, 2022, doi:10.3390/jcm11030497_

Round 1

Reviewer 1 Report

he authors analyzed 198 MBIC patients treated with cystectomy followed by platinum-based adjuvant chemotherapy recruited by four hospitals. They investigated and correlated clinicopathological patient characteristics and survival with the immunohistochemical expression of CD36, a critical player in lipid metabolism. By using a non-parametric approach, they established that CD36 immunopositivity was significantly associated with more aggressive phenotypes highlighting association with advanced stages and greater lymph node involvement.

In my opinion, the present study is not particularly original, but the research question and the design is appropriate, as well as the impact of CD36 IHC expression across a multistitutional cohort deserves relevance. There is an urgent need to bring affordable biomarkers for better prognostication in bladder cancer, not only in patient who underwent radical cystectomy upon MBIC but also in NMIBC patients not responsive to immunotherapy, please cite a recent review New Roadmaps for Non-muscle-invasive Bladder Cancer With Unfavorable Prognosis doi: 10.3389/fchem.2020.00600. T

Moreover, the CD36 expression has recently been evaluated in paraffin-embedded tumor and para-tumor tissues by immunohistochemistry across distinct tumor types. The authors  performed a pan-cancer analysis. So, I would suggest to include in the discussion more critical judgement of the findings in the light of the data recently reported here Prognostic and immunological role of CD36: A pan-cancer analysis  doi: 10.7150/jca.50502.

Line 148: I would better understand … Did you get ambiguous IHC results for 9 patients ? Please add more clarifications.

Line 152: Fig.1 unclear, may be improved. I would add more details.

257 MBIC recruited patients à How many MBIC patients did you recruit for each center?

198 eligible patients à How many were eligible for the final analysis for each center?

About the legend, I suggest “Flow chart of the patients included in this study”.

Line 231: add references.

Line 240: Table 2 footnotes please edit * Statistical significative p value ≥ 0.05. ** Statistical significative p value ≥ 0.01. I guess p values should be <, or <=.

Reviewer 2 Report

The authors analyzed the tissue of 198 patients undergoing radical cystectomy for expression of CD36. Thereby, they aim to evaluate the prognostic impact of CD36 immunohistochemical expression in patients with muscle-invasive bladder cancer treated with cyctectomy and adjuvant chemotherapy. While this setting and hypothesis are scientifically and clinically interesting/relevant, the study holds several issues that have to be addressed:

  • CD36 is described as a glycoprotein that has a role in metabolic pathways. Therefore, it would be interesting to include BMI of patients into the patient characteristic and evaluate a potential correlation with CD36 expression.
  • The majority of patients is lymph node positive. Is there any data describing expression of CD36 in those lymph nodes? Is there any correlation with the primary tissue
  • Conclusion: Is this data enough to conclude that CD36 is an important biomarker in MBIC.
